# How urban form impacts flooding

Sarah K. Balaian[1], Brett F. Sanders [1,2] ✉ &
Mohammad Javad Abdolhosseini Qomi [1,3] ✉

Urbanization and climate change are contributing to severe flooding globally, damaging infrastructure, disrupting economies, and undermining human well-being. Approaches to make cities more resilient to floods are emerging, notably with the design of flood-resilient structures, but relatively little is known about the role of urban form and its complexity in the concentration of flooding. We leverage statistical mechanics to reduce the complexity of urban flooding and develop a mean-flow theory that relates flood hazards to urban form characterized by the ground slope, urban porosity, and the Mermin order parameter which measures symmetry in building arrangements. The mean-flow theory presents a dimensionless flood depth that scales linearly with the urban porosity and the order parameter, with different scaling for disordered square- and hexagon-like forms. A universal scaling is obtained by introducing an effective mean chord length representative of the unobstructed downslope travel distance for flood water, yielding an analytical model for neighborhood-scale flood hazards globally. The proposed mean-flow theory is applied to probe city-to-city variations in flood hazards, and shows promising results linking recorded flood losses to urban form and observed rainfall extremes.

Flooding is one of the most frequent and costliest natural disasters[1,2], and impacts are often concentrated in cities[3,4]. The severity of flooding is increasing from more intense storms driven by a warming climate[5], expanding development into hazardous areas[6], and high levels of physical and social vulnerability[3,7]. Vulnerabilities include building and infrastructure designs that do not tolerate exposure to rising and fast-moving flood waters[8], economies that are hindered by infrastructure impacts and job losses[9], and flooding that disproportionately exposes disadvantaged and marginalized communities[7]. In a future marked by more extreme weather events and populations concentrated in cities, attention to the physical form of cities and their vulnerability to flood impacts is of growing importance. This includes the development and application of climate-resilient architectural design approaches[10,11], so communities and households can significantly limit the severity of flood losses and more readily bounce back from severe events. But this also calls for attention to the urban form of the community, including the density of development and the configuration of the street network and its relation to ground slopes which drive the movement of flood water.

Research to date suggests that the configuration of urban forms has the potential to alter the patterns and severity of flooding substantially[12,13], but an understanding of the role of urban forms, globally, in shaping flood impacts is limited. Indeed, urban planning initiatives are underway in cities around the world to adapt at-risk communities to more intense flooding and other climate risks, meet needs for housing and commerce, and manage ecosystem health and the provision of ecosystem services; these would benefit from foundational knowledge–a mean-flow theory–describing the effect of urban form on the concentration of flood severity. That is, the goal here is to relate how the depth and velocity of urban flooding at neighborhood scales relate to the orientation and configuration of street layouts and buildings, given the volumetric flow rate. When combined with data and models estimating volumetric flow rates in cities, for example, from heavy rainfall, stormwater surcharge, and overbank streamflows, a mean-flow theory could aid analyses of factors driving present-day flooding hotspots across cities and inform future urban planning with respect to flood risk. Such mean-flow theory is, in principle, akin to the empirical Manning equation for the flow

[1]Department of Civil and Environmental Engineering, University of California, Irvine, Irvine, CA, USA. [2]Department of Urban Planning and Public Policy, University of California, Irvine, Irvine, CA, USA. [3]Department of Materials Science and Engineering, University of California, Irvine, Irvine, CA, USA. ✉ e-mail: bsanders@uci.edu; mjaq@uci.edu

properties of streams[14], $Q = AR^{2/3}\sqrt{\alpha}/n_M$, where flow rate $Q$ is related to Manning's roughness $n_M$, flow cross-section and hydraulic radius $A$ and $R$, and the bottom slope $\alpha$, respectively. The Manning equation and other empirical models (e.g., Chezy) have proven instrumental and robust for designing flood channels, and here we seek an equivalent for urban flood plains.

Our inspiration for an urban flooding mean-flow theory is taken from how complex systems are approached in statistical mechanics. Urban forms epitomize a complex system much akin to granular media[15], disordered porous solids[16], glassy systems[17], and complex fluids[18], to name only a few. When viewed deterministically, flooding through a complex urban form depends on the location of each obstruction. However, in an average sense, the flow behavior should be governed by a few characteristic state variables reflective of the urban form, e.g., building density, and indicators of how clusters of buildings are configured. This averaging is motivated by Boltzmann's treatise that proves the equilibrium properties of non-interacting classical particles remain independent of their location and instead rely on the system's volume—a fundamental thermodynamic state variable[19]. To reduce the complexity of urban flooding, we need a large ensemble of refined flow observations to build our mean-flow theory. Experimental studies[20] and field measurements[21] have proven insightful when studying the role of obstructions on urban flooding, yet to date, they remain strictly limited to extremes of simplified obstruction arrangements or case-specific urban forms. Hence, computational modeling is the only viable approach to collect statistically significant data on urban flooding dynamics over a representative spectrum of urban forms.

The flow structure and boundary layers of urban flooding can be simulated with Reynolds-averaged Navier-Stokes, large-eddy simulation, and direct numerical simulation[22]. However, in pursuit of computational efficacy, we rely on a shallow-water hydrodynamic model[23] and produce thousands of fine-resolution, building-resolved simulations for different realizations of urban form. Shallow-water theory builds on the assumption of hydrostatic pressure and nearly horizontal flow, and shallow-water models have been validated at neighborhood and larger scales by extensive laboratory and field studies and urban flood modeling[20,24–29]. While urban flooding can involve high levels of

complexity, such as flows into basements, subway systems, and usually involves a mix of underground flows through storm pipes and overland flow along streets[30], we are specifically interested here in the depth and velocity of overland flow through cities caused by the amount of overland flow in response to hazard drivers (e.g., rainfall, overbank flows, stormwater surcharge). Hence, our simulation approach involves measuring the steady-state depth and velocity of flooding for a given flow rate under a wide range of urban forms.

In this work, we assemble an ensemble of urban flood simulations and reduce the complexity of urban flooding through dimensional analysis of simulation results and consideration of conservation laws into a master curve that constitutes a mean-flow theory. This theory establishes a mathematical equation that relates flooding attributes—notably depth and intensity defined as the product of flood depth and velocity—to the ground slope, urban form, and flow rate encountered across cities. Moreover, we seek a model that explains the global variability of urban flooding hazards at neighborhood and larger scales and provides planning-level support for more flood-resilient cities.

## Results and discussion
### Mean-flow theory for urban flooding

City building arrangements are highly varied globally, as indicated in Fig. 1 (Supplementary Note I and Materials and "Methods" Section A). For example, the alignment of buildings in Chicago, USA (Fig. 1c) is structured, while Lagos, Nigeria is unstructured (Fig. 1h). A mean-flow theory of urban flooding applicable to global variations in building layouts (e.g., Fig. 1(a–h)) calls for a systematic parameterization of urban form, and here we draw from studies of the physics of phase transition and disordered materials[31,32], where the Mermin order parameter[33] ($\chi$) has been used to quantify the spatial degree of order/disorder of a system of particles. At the scale of individual particles (or buildings), the Mermin order parameter determines the symmetry of $c_n$ nearest neighbors to the reference particle[34]. For instance, for building arrangements with square symmetry, $c_n = 4$, and those with hexagonal symmetry feature $c_n = 6$. Furthermore, the local order parameter can be averaged to extract a collective disorder attribute representative of an urban region, as

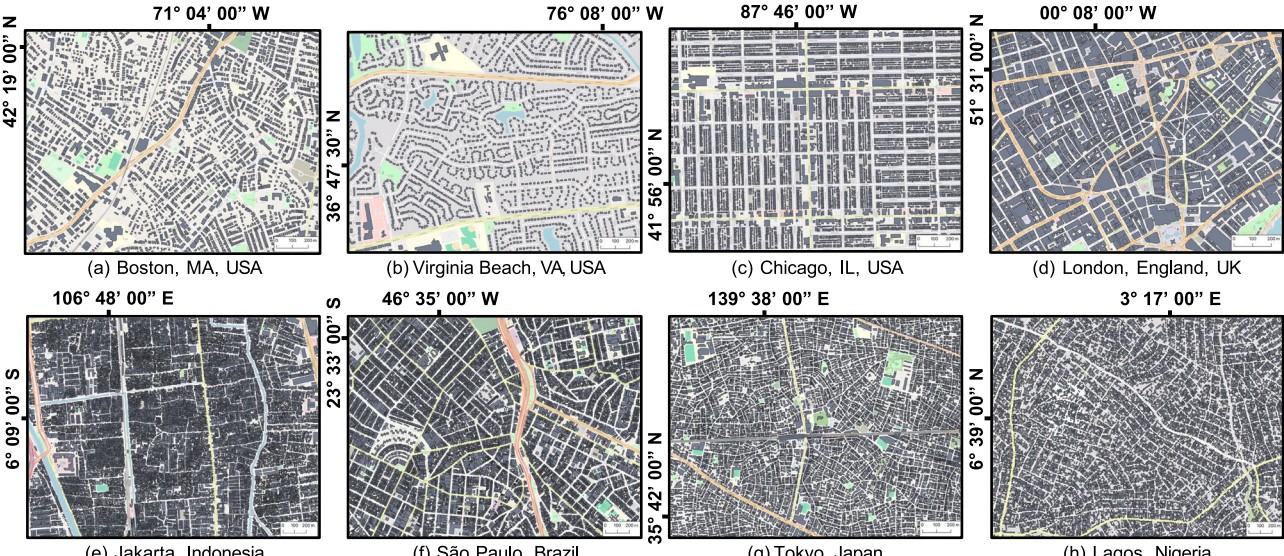

**Fig. 1 | Urban form across the cities in North and South America, Europe, Africa, and Asia is highly varied.** Selected areas from (**a**) Boston, (**b**) Virginia Beach, (**c**) Chicago, (**d**) London, (**e**) Jakarta, (**f**) Sao Paulo, (**g**) Tokyo, and (**h**) Lagos. The building footprints, streets, highways, parks, and rivers are displayed in gray, yellow, orange, green, and blue, respectively. The coordinates of each city are defined along the top and left sides of the panel. Basemaps provided by OpenStreetMap (https://www.openstreetmap.org/copyright) used in conjunction with Microsoft's Building Footprints and Google's Open Buildings, data made available via an Open Database License (https://opendatacommons.org/licenses/odbl/).

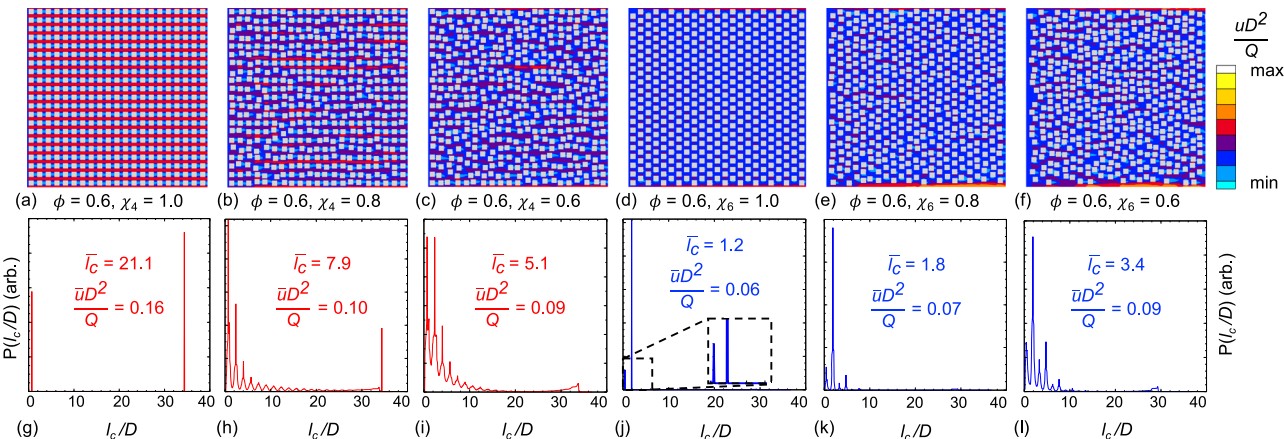

**Fig. 2 | The qualitative study of flow pathways in synthetic urban forms shows the impact of urban porosity, order, and chord length on flood velocity distribution.** Non-dimensionalized flood velocity maps for unit cells of urban forms with (**a**–**c**) square and (**d**–**f**) hexagon-like patterns with porosity $\phi$=0.6 and Mermin order $\chi$=1.0, 0.8, and 0.6, respectively. The color bar on the right represents the velocity scale for all velocity maps. Gray squares represent flow obstructions. While the unit cell $l_u$ is ~33$D$, the channel length is set to 20$l_u$ to mitigate size effects. The normalized chord length distribution $P(l_c')$ for each urban form is placed below each velocity map in panels (**g**–**l**); see the text for the definition.

follows:

$$\chi_{c_n} = \frac{1}{c_n N} \sum_{j=1}^{N} \sum_{k=1}^{c_n} \exp(ic_n\theta_{kjk_0}) \tag{1}$$

where $\theta_{kjk_0}$ is the angle between the central building $j$ and its neighbors $k$ and $k_0$, where $k_0$ is a fixed neighbor and $k$ permutes over all possible neighbors of the $j$th building. When $\chi_{c_n}$ is greater than 0.9, the synthetic urban form exhibits a pronounced symmetry and resembles a planned urban layout, e.g., Chicago, Fig. 1c. As disorder increases, the symmetry diminishes, and the synthetic urban form features characteristics of an organically nucleated and grown urban environment, e.g., Jakarta, Fig. 1e. Here, we adopt the order parameter concept to quantitatively characterize the disorder of buildings within a city alongside the urban porosity $\phi$, which characterizes the amount of pore space and is inversely related to the building density.

Ensembles of synthetic city urban forms at a given porosity and disorder are developed using hybrid reverse Monte Carlo[35,36] (HRMC); see *Materials and "Methods"* Section B and Supplementary Notes II–III for detail. Furthermore, to support shallow-water modeling and specifically the characterization of steady-state, uniform flow conditions, a rectangular urban domain is created with a length ($L$), width ($W$), longitudinal slope ($\alpha$), and a Manning coefficient for bottom resistance ($n_M$), and the domain is populated with $N$ square buildings of side length ($D$). Due to HRMC's computational cost, we only generate building patterns in a unit cell of length $l_u$ and repeat the unit cell to cover the entire channel length. The chosen level of disorder controls the irregularity of building locations (see Supplementary Fig. S1), and the urban porosity follows as $\phi = 1 - ND^2/(WL)$. To characterize flooding, the shallow-water equations are solved on a fine-resolution, unstructured computational mesh (ParBreZo), where mesh edges are aligned with building walls to enforce a no-flow boundary condition[23,28], see *Materials and "Methods"* Section C, Supplementary Figs. S2–S3.

Our simulations resolve flooding within the pore space between buildings, and solutions reflect the bulk effect of building form drag. Moreover, using a Godunov-type finite volume scheme for solving the shallow-water equations allows us to resolve supercritical flows, subcritical flows, and flows with hydraulic jumps[23]. Solutions include localized super-critical flows ($Fr > 1$) along steep slopes, but average flows across neighborhoods are found to be largely sub-critical ($Fr < 1$).

As has been observed in experiments[37], simulations show that a hydraulic jump occurs when localized supercritical flows encounter obstructions[37].

Figure 2a–f show the color map of non-dimensionalized flood velocity in the downslope direction ($uD^2/Q$) as a function of flow velocity ($u$), flow rate ($Q$), and building size ($D$), for neighborhoods of varying porosity and disorder. Relative to square symmetry at $\phi = 0.6$, we observe the choking of flow with decreasing $\chi_4$. This indicates that increasing disorder restricts the flood flow in urban forms that possess a near square symmetry (regular layout), but has an opposite effect in synthetic urban models with nearly hexagonal symmetry (or a staggered layout) representative of flow paths along diagonal directions. In particular, at $\phi = 0.6$, the disorder opens new flood flow pathways through clusters of buildings much akin to flow pathways through disordered granular media[38].

These qualitative observations suggest that both the symmetry and irregularity of street grids in cities contribute to the tortuosity and constriction of flow paths at neighborhood and larger scales, which in turn shape the distribution of flood depths and velocities responsible for safety risks and losses[39]. To characterize how the symmetry and irregularity interact with ground slope to control the neighborhood-scale flood levels, we leverage the non-dimensionalized chord length distribution $P(l_c'(\theta))$, which represents the distribution of randomly placed line segments fitting within pore space between two building walls[40] in all directions ($\theta$), see Supplementary Note IV and Supplementary Fig. S4 for detail. For simplicity, we here focus on chord length in the direction of the steepest descent. The chord length represents the distance over which a fluid particle can accelerate under the influence of gravity and bottom shear before being blocked by an obstruction, and Fig. 2(g–l) show the distribution of $P(l_c')$ for the building patterns shown in Fig. 2(a–f), respectively. The floodplain with perfect square symmetry exhibits a bimodal distribution of long and short chord lengths corresponding to the unit cell's length and nearest neighbor distance. By introducing disorder, additional peaks appear at intermediate values of chord length. The perfectly hexagonal floodplain also displays a bimodal distribution and behaves similarly; however, the large chord length corresponds to the second nearest neighbor. A representative chord length for any urban form follows with the expectation of $l_c'$ excluding the first and second nearest neighbor chords that do not effectively contribute to the flood drainage due to stagnation effects. Focusing on chord lengths beyond

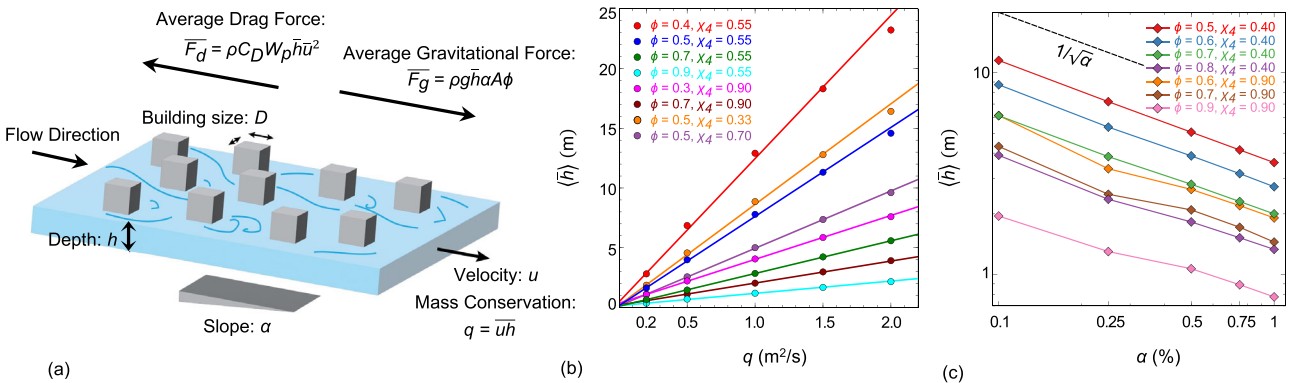

**Fig. 3 | Considering mass and momentum conservation laws to develop a non-dimensionalized relation between flood height, driving forces, and urban form attributes.** **a** A schematic of the control volume through which water, shown in blue, flows between gray square obstacles. In dense urban environments, the floor shear is negligible compared to the other forces, and the gravitational force balances the form drag at the steady state. A limited number of one-factor-at-a-time flow simulations corroborate with the proposed dimensionless relationship, showing the flood height (**b**) scales linearly with flow rate per width (*q*), and (**c**) is proportional to the inverse square root of the bottom slope ($\alpha^{-1/2}$).

three building sizes, we define the normalized average effective chord length as:

$$\bar{l}_c = \int_3^{l_u/D} l_c' P(l_c') d(l_c') \tag{2}$$

where $l_c' = l_c/D$ and $l_u$ is the length of the unit cell. The corresponding $\bar{l}_c$ and average non-dimensionalized velocity for each distribution also appear in Fig. 2g–l. We observe qualitatively that the average flood velocity is directly correlated with $\bar{l}_c$, meaning that long $\bar{l}_c$ results in higher average velocities regardless of the underlying urban symmetry.

To reduce the observed complexity in urban form-induced flood behavior, we consider the mass and momentum conservation laws for dense urban forms, where the bottom shear is negligible compared to the distributed form drag. In a spatially large enough control volume that sufficiently represents the flow heterogeneity, the average gravitational force ($\bar{F}_g = g\bar{h}A\phi\cos(\alpha)\sin(\alpha)$) balances the resultant form drag ($\bar{F}_D = C_D A_p \overline{u|V|}$) in the steady-state limit, Fig. 3(a). In this case, while gravity (*g*) drives the flow with average vertical height ($\bar{h}$) through available urban pore space ($A\phi$), the drag resists the flow. This resistive force is proportional to the drag coefficient ($C_D$), the so-called projected area ($A_p = \bar{h}W_p\cos(\alpha)$)[41], and spatially averaged velocity squared component ($\overline{u|V|} = \overline{u|\sqrt{u^2 + v^2}|}$). The average perpendicular component of flow velocity (*v*) is zero, and its magnitude should be smaller than *u*, meaning $\overline{u|V|} \approx \overline{u^2}$. Invoking mass conservation ($q = \bar{u}h\phi\cos(\alpha)$), we can eliminate the drag dependence on velocity and estimate this force using average height, $\bar{F}_d \approx C_D W_p q^2/(\bar{h}\phi^2\cos(\alpha))$. Therefore, through force balance ($\bar{F}_g = \bar{F}_d$), we can estimate the average flood height as follows,

$$\frac{\bar{h}\sqrt{gD}}{q} = \sqrt{\frac{C_D W_p D}{\cos^2(\alpha)\sin(\alpha)A\phi^3}} = \frac{\Pi\sqrt{C_D}}{\cos(\alpha)\sqrt{\sin(\alpha)}} \approx \frac{\Pi\sqrt{C_D}}{\sqrt{\alpha}} \tag{3}$$

The main challenge in determining the average flood height pertains to the exact measurement of projected width ($W_p$), an extensive quantity that depends on urban form. Therefore, $A\phi^3/W_p$ can be viewed as a characteristic length scale whose size is proportional to the building size, $A\phi^3/W_p = D/\Pi^2$, where similar to $C_D(\frac{r_i}{D})$, $\Pi(\frac{r_i}{D})$ is a dimensionless function of the building arrangements ($r_i$) in the urban layout. For the maximum 10% slope considered in this study, the approximation of $\cos(\alpha)\sqrt{\sin(\alpha)} \propto \sqrt{\alpha}$ introduces less than 1% error.

The average flood height in Eq. (3) is deterministic if the locations of all buildings are known. However, when viewed through statistical mechanics, the flood height should, in principle, be only dependent on a small number of statistical attributes of the floodplain rather than the unique locations of all buildings. While this is an unconventional concept in the context of urban flooding, it is a well-established concept for studying the thermodynamics of granular media and glassy systems[17]. In this sense, the ensemble-averaged dimensionless flood height can be thought of as a hydrodynamic state function of the city that depends on hydrodynamic state variables $\phi$, $\chi$, and $\bar{l}_c$, which yields the following dimensionless relation:

$$\frac{\langle\bar{h}\rangle\sqrt{gD}}{q} = \frac{1}{\sqrt{\alpha}} C_D'(\phi, \chi, \bar{l}_c) \tag{4}$$

where $\langle.\rangle$ denotes the ensemble average and modified drag coefficient $C_D' = \langle\Pi\sqrt{C_D}\rangle$. Interestingly, we derive the same relation through considerations of dimensional homogeneity, see Supplementary Note V. Regardless of the derivation method, the theory suggests the ensemble-averaged dimensionless flood height ($\langle\bar{h}\rangle\sqrt{gD}/q$) should depend on dimensionless geometric factors, the bottom slope, and the drag coefficient that depends on the porosity, order parameter, and average chord length in the flow direction. To test this theory, we perform one-factor-at-a-time flow simulations varying slope and flow rate per width for different urban forms with various porosity and order parameters, Fig. 3(b–c). Corroborating with the proposed theory, the numerical simulations show $\langle\bar{h}\rangle$ scales linearly with *q* and is inversely proportional to the square root of the bottom slope, $1/\sqrt{\alpha}$. We note that while our theory and Manning equation show $\langle\bar{h}\rangle \propto 1/\sqrt{\alpha}$, they differ in predicting the exponent in $\langle\bar{h}\rangle \propto q^m$ scaling relation. In an empty wide rectangular channel, where flow is driven by gravity and bottom shear, the Manning equation predicts $\langle\bar{h}\rangle \propto q^{3/5}$. However, as discussed before, in a densely-packed urban flood channel, the form drag balances the gravity leading to a new scaling relation, i.e., $\langle\bar{h}\rangle \propto q^1$. Therefore, the proposed change in exponent *m* is intimately related to the change in underlying physical processes governing momentum transfer.

To further verify the proposed theoretical framework, we need to demonstrate that $C_D'(\phi, \chi, \bar{l}_c)$ can solely explain the distributions of flood depth over a range of flow simulations with varying $0.4 \leq \phi \leq 1$, $0.3 \leq \chi_4, \chi_6 \leq 1$, $\alpha$, and *q*. In an attempt to collapse the results, we first neglect the impact of urban order, i.e., $C_D' = f_1(\phi) \propto (1 + a_{c_n}\phi)^{s_{c_n}}/\phi^{p_{c_n}}$, where $a_{c_n}$, $s_{c_n}$ and $q_{c_n}$ are specific to square and hexagonal symmetries as denoted by the

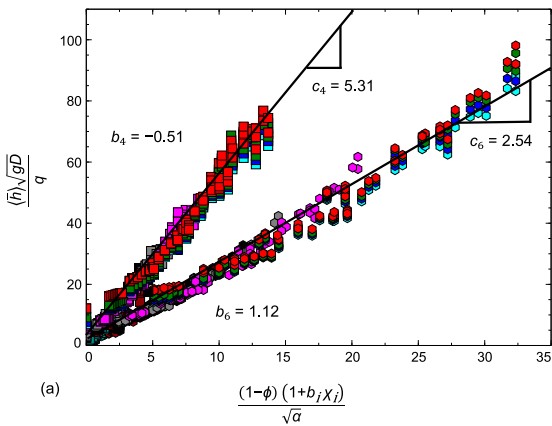

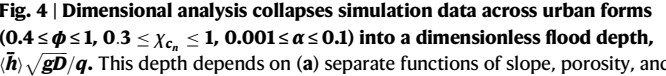

(a)                                                      (b)

**Fig. 4 | Dimensional analysis collapses simulation data across urban forms** ($0.4 \leq \phi \leq 1$, $0.3 \leq \chi_{c_n} \leq 1$, $0.001 \leq \alpha \leq 0.1$) **into a dimensionless flood depth,** $\langle \bar{h} \rangle \sqrt{gD}/q$. **This depth depends on (a)** separate functions of slope, porosity, and

order for rectangular and hexagonal symmetry and **(b)** a single function of slope, porosity, and average chord length.

coordination number $c_n$. This approach reduces the search for the state function to an optimization problem. Regardless of the underlying symmetry, $s_{c_n} \approx 1$, $p_{c_n} \approx 0$ and $a_{c_n}$ converges to -1, meaning $f_1 \approx (1 - \phi)$, better known as the packing fraction in the physics of granular media. This parameter mirrors the urban building coverage, which is suggested to be the most influential factor affecting flood depth via a sensitivity analysis[12]. The coverage parameter also appears in flow through channels covered by vegetation and is shown to scale flow resistance[42]. However, the packing fraction collapses data only at high porosity and becomes increasingly insufficient in dense textures, where the disorder becomes a controlling factor, see Supplementary Fig. S8a.

To account for the disorder effects at low porosity, we assume $C'_D = f_1(\phi) \times f_2(\chi)$. For simplicity, we focus on linear functions through the Taylor expansion, $C'_D \approx c_{c_n}(1 - \phi)(1 + b_{c_n}\chi_{c_n})$, where $b_{c_n}$ and $c_{c_n}$ are fitting parameters. In particular, we arrive at the following relation:

$$\frac{\langle \bar{h} \rangle \sqrt{gD}}{q} = c_{c_n} \frac{(1 - \phi)(1 + b_{c_n}\chi_{c_n})}{\sqrt{\alpha}} \quad (5)$$

where slopes $c_4$ and $c_6$ are 5.31 and 2.54, and $b_4$ and $b_6$ are -0.5 and 1.1, respectively, as noted in Fig. 4a. Interestingly, we note $c_4/c_6 \approx 2$ and $b_4/b_6 \approx -1/2$.

As qualitatively observed in Fig. 2, the introduction of disorder into square and hexagonal urban forms has an opposite effect on flood behavior. This is now quantitatively confirmed by the change in the $b_{c_n}$ sign. We also note the proposed linear relationships do not capture the flood depth for floodplains without obstructions. The relevant dimensionless quantity in the limit of high $\phi$ is $\langle \bar{h} \rangle^{5/3}/(q n_M)$, which is confirmed by the convergence of simulation data when $(1 - \phi)(1 + b_{c_n}\chi_{c_n})/\sqrt{\alpha}$ tends to zero, see Supplementary Fig. S8b.

We can now examine the impact of urban form on flood depth in synthetic city neighborhoods. Leveraging Eq. (5) at constant porosity, the flood depth ratio between square- and hexagon-like patterns becomes $c_4(1 + b_4\chi_4)/(c_6(1 + b_6\chi_6))$. This ratio is $\approx 1/2$ for perfectly ordered cities, meaning flood depth is twice as severe in hexagonal cities. As the disorder increases to the point that the underlying patterns become indistinguishable, i.e., $\chi_4 = \chi_6 = 0.5$, the ratio becomes $\approx 1$, and as expected, there is no difference in flood depth. In this scenario, the disorder increases flood depth by $\approx 1/2$ in a square urban pattern, while it decreases the flood depth by $\approx 1/4$ in hexagonal form.

The inherent drawback of Eq. (5) as a mean-flow theory is that it does not provide a universal relation between dimensionless quantities due to dependence on the underlying symmetry. As noted in Fig. 2, the flood depth is associated with open flow pathways characterized by $\bar{l}_c$. This prompts proposing $C'_D(\phi, \bar{l}_c) = f_1(\phi) \times f_3(\bar{l}_c)$. Focusing on an $f_3$ function of the form $\bar{l}_c^\beta$, we find through optimization that $\beta \approx -1/2$. As shown in Fig. 4b, this heuristic approach collapses the bifurcated results into a linear relationship as follows:

$$\frac{\langle \bar{h} \rangle \sqrt{gD}}{q} = d_1 \frac{(1 - \phi)/\sqrt{\bar{l}_c} + d_0}{\sqrt{\alpha}} \quad (6)$$

where $d_0 = 0.07$ and $d_1 = 5.62$. Eqs. (5)–(6) are valid in the range of $0.4 \leq \phi$ considered here. Eq. (5) becomes increasingly inaccurate at lower porosity as flood depth should increase non-asymptotically at $\phi = 0$. Eq. (6), however, becomes singular at $\phi = 0$, since $\bar{l}_c$ tends to zero in this regime. This equation constitutes a mean-flow theory of urban flooding that relates the average flood depth to urban form attributes. Note that increasing slopes and increasing chord length act to decrease flood depth due to increases in flood velocity. Hence, the mean-flow theory predicts shallow, high-velocity flows along streets oriented in the downslope direction, an especially dangerous flow regime that has been described as ultrahazardous flooding[43,44].

### Global Flood Hazard Distribution based on the Mean-Flow Theory

The proposed mean-flow theory is now applied to examine the city-to-city variability of urban flooding hazards; see *Materials and "Methods"* Section D. Selecting twenty cities across five continents, we analyze urban porosity and chord length in the direction of the steepest slope at a km² grid resolution as shown in Fig. 5(a–b). Similar to the synthetic floodplains, we observe a correlation between urban porosity and normalized average chord length, Fig. 5c. The denser cities feature smaller chord lengths, as evident in San Francisco and Lagos. However, a notable outlier to this trend is São Paulo, where long streets in densely developed neighborhoods create long corridors for flood passage. Loosely-packed cities such as Virginia Beach also exhibit longer chord lengths due to the sparsity of flow obstructions.

To characterize cities in terms of their flood hazards, we define two dimensionless flood hazard indicators: flood depth and intensity; see Supplementary Note VI. For the dimensionless flood depth hazard $H'$, we use our mean-flow theory in Eq. (6), which is independent of the underlying symmetry and more applicable to real-world complex urban forms. Flow rates required by the mean-flow theory, $q$, are

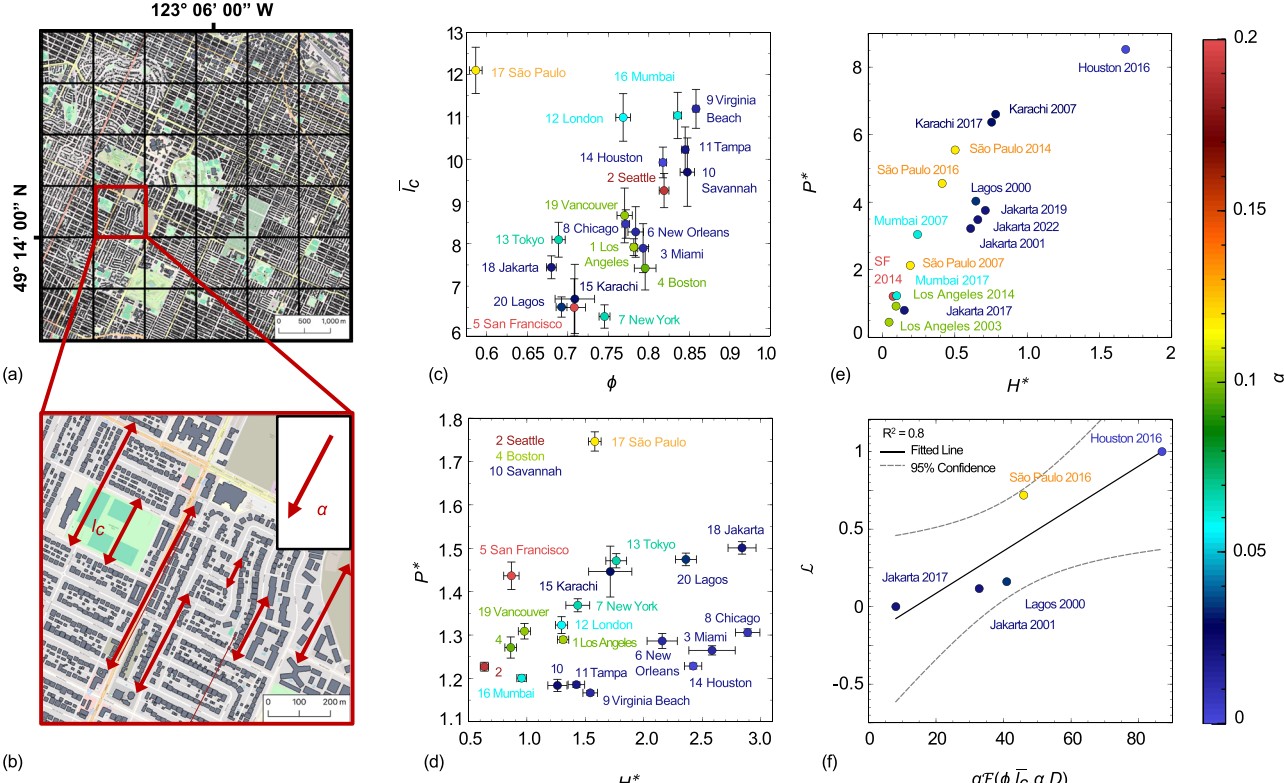

**Fig. 5 | The application of mean-flow theory to probe flood hazard globally.**
Flood hazards across cities worldwide are examined by (**a**) Subdividing cities into 1 km x 1 km tiles (shown in Vancouver, BC, Canada) and evaluating the topographic slope $\alpha$, porosity $\phi$, and order $\chi$ and (**b**) estimating the average chord length in the direction of descent to reveal (**c**) the porosity and chord length across 20 twenty cities; error bars show the margin of error with 90 percent confidence. Application of the mean-flow theory reveals: (**d**) the flow rate-normalized flood inundation $H^* = \langle \bar{h} \rangle / h_{ref}$ and non-dimensionalized flood intensity $P^* = \langle \overline{hu} \rangle / hu_{ref}$ for twenty cities with constant flow rate of $Q = 1 m^3/s$, (**e**) the theoretical flow rate-normalized flood inundation $H'$ and non-dimensionalized flood intensity $P'$ for recorded flood events. Finally, (f) The normalized monetary damages $\mathcal{L}$ are compared to the product of the inflow rate $q$ and urban form factor $\mathcal{F}(\phi, l_c, \alpha, D)$ among different flash flooding events. In panels (c-f), the markers' colors represent the average bottom slope of each city defined by the color bar to the right side. Building footprint data for Karachi and Mumbai may not be sufficient. Basemaps provided by OpenStreetMap (https://www.openstreetmap.org/copyright) used in conjunction with Microsoft's Building Footprints and Google's Open Buildings, data made available via an Open Database License (https://opendatacommons.org/licenses/odbl/).

estimated as the product of an extreme event precipitation intensity and a length scale for runoff accumulation, taken here to be 1 km. We then divide $\langle \bar{h} \rangle$ by $h_{ref}$, the reference flood depth in an obstruction-free channel with $\alpha_{ref} = 0.1$ and $q_{ref} = 0.001$ m²/s, which are respectively representative of average slopes and precipitation rates observed in our twenty cities. We also define the dimensionless flood intensity as the average product of flood depth and velocity normalized by the flood intensity in an obstruction-free channel inundated by $q_{ref}$, $P^* = \langle \overline{hu} \rangle / (\overline{hu})_{ref}$. This dimensionless quantity is controlled by the inflow rate at the steady state, meaning $P^* = q/(q_{ref}\phi)$. This indicates the normalized flood intensity at the urban scale is independent of its form and bottom slope, and controlled by porosity and flow rate only; see Supplementary Fig. S9.

How susceptible are cities to urban flood hazards based on urban form alone? We can now address this question by computing the dimensionless flood depth and intensity given the urban porosity, chord length, and slope for each city, as shown in Fig. 5d. Note that with $q$ computed with a 1 km runoff accumulation length scale for all cities, differences in flood depth and intensity are completely attributable to differences in urban form and precipitation intensity, and not to differences arising from larger scale drainage patterns or urban drainage infrastructure that contribute to the accumulation of urban flood flows. Most cities are clustered toward the low-hazard region, i.e., low flood depth and intensity. Chicago's urban form is conducive to high flood depth due to its relatively low porosity and flat terrain. San

Francisco faces the opposite hazard, where the flood intensity overshadows flood depth due to the steep topography and low urban porosity. High-hazard cities are marked by high levels of flood depth and intensity and include Tokyo, Lagos, Jakarta, Karachi, and São Paulo. Compared to other cities in this study, these cities exhibit a very low porosity and mild slope. In the case of São Paulo, there are also long and steep flow pathways characterized by high chord length and bottom slope.

We reviewed the Emergency Events Database (EM-DAT)[45] for extreme events that occurred within the 20 cities considered in this study and used archives of historical precipitation and a simple rainfall-runoff approximation to estimate the inflow rate, $q$, see *Materials and "Methods"* Section D. Figure 5(e) maps hazard severity for sixteen extreme events classified as flash floods using our dimensionless flood depth and intensity. São Paulo, Jakarta, Karachi, Toyko, and Houston emerge as the cities that have experienced the most intense flash flood hazards, based on dimensionless flood depth, intensity, or a combination of the two. Moreover, the occurrence of multiple events in São Paulo, Jakarta, and Karachi suggests that these cities are amongst the most hazardous cities in the world for flash floods based on a combination of extreme event frequency and urban form.

Validating the proposed mean-flow theory for describing city-to-city variations in urban flooding is extremely difficult because extreme event flooding of street grids is not systematically documented with ground-based sensors and satellite-based remote sensing methods for

flood inundation do not perform well in urban areas[46]. EM-DAT includes metrics of economic losses, which we expect to be associated with precipitation, flood depths, and flood velocity. However, flood losses depend on many physical and social vulnerabilities in addition to hazard severity. Thus any attempt to link hazard severity to economic impacts is expected to be highly uncertain. We focused on five flash flooding events from EM-DAT with economic loss data within the twenty selected cities of this study. Due to the economic variability of the impacted cities, we normalize the reported damages by the respective country's gross domestic product and re-scale the damage values between zero and unity. Figure 5(f) reveals a correlation between the normalized monetary damages, $\mathcal{L}$, and a linear combination of flood hazard indicators as follows:

$$\mathcal{L} \propto H^* + \kappa P^* = q\mathcal{F}(\phi, l_c, \alpha, D) \tag{7}$$

where

$$\mathcal{F}(\phi, l_c, \alpha, D) = \frac{C'_D(\phi, l_c)}{h_{ref}\sqrt{gD\alpha}} + \frac{\kappa}{q_{ref}\phi} \tag{8}$$

This relationship provides a rather intuitive yet insightful finding that flood damages are linearly related to $q$ and an urban form factor, $\mathcal{F}$. The positive correlation between $\mathcal{L}$ and $F$ is in general alignment with established theories and models for flood damage, including widely used depth-damage curves [e.g.,[47]]. If we set $\kappa = 0$ in our damage model, Eq. (8), it is simplified to a linear depth-damage curve. However, we find that the reported EM-DAT damages cannot be solely explained by flood height, which necessitates the consideration of flood intensity as a damage-driving factor.

The Houston (2016) event appearing in Fig. 5f was the most costly of the flash flood events recorded across the cities examined for this study. However, the most costly extreme events in recorded history were all classified by EM-DAT as tropical cyclones, not flash floods, and the top six events as of 2019 all occurred in the U.S.[2]. When we compared tropical cyclone losses and our estimates of urban flood hazards from rainfall, we did not find a correlation. Indeed, damage from tropical storms results from several hazard drivers, including wind, storm surge, and rainfall[2].

We acknowledge several limitations of our approach, including the representation of urban forms with arrays of square obstructions as opposed to more organic forms, assumed uniformity in the runoff accumulation length scale, the approximation of urban flooding as a two-dimensional flow without fully resolving the turbulent boundary layers that contribute to form drag, and the limited number of cities sampled for the development of our mean-flow theory. Additional research could help with understanding the importance of each of these methodological factors relative to the development of a mean-flow theory for urban flooding. Nevertheless, given the range of urban porosity, order values, and floodplain slopes considered herein, our approach systematically considers an immense spectrum of urban forms. Indeed, we successfully reduce the complexity of urban flood flow data into a master curve that provides an intuitive explanation of the distribution of flood hazards globally across cities, and that aligns with observations of flood losses.

The mean-flow theory presented here thus offers a foundational approach for examining how urban form affects the physical vulnerability of cities to flooding. Application of the mean-flow theory at neighborhood and larger scales could help to understand why specific cities, or even neighborhoods within cities, are most susceptible to flood impacts. The mean-flow theory could also support the planning of new developments in anticipation of flood exposure, and the development of flood resilience plans and programs. Hence, the mean-flow theory offers a promising entry point for urban planning of safer and more resilient cities.

## Methods

### Urban form data
We assimilate publicly available building footprints, digital elevation models (DEMs), and city boundary data sets for twenty cities worldwide, see Supplementary Notes I. For cities in the United States, building footprint data is obtained through Microsoft Maps' nationwide open building footprints data set[48]. For most cities outside the United States, we extract building footprints from OpenStreetMap[49], a publicly available crowd-sourced database. To supplement this database, we also obtain building footprints for African and Southeast Asian countries through Open Buildings, a building footprint database developed through deep learning of satellite imagery[50]. We obtain the elevation data from MERIT[51], a high-accuracy global digital elevation model at roughly 90-meter resolution. While the MERIT DEM was found to contain an RMSE of ~4m[52], this error translates to a ± 0.008 change in slope and is likely present in all cities, having little effect on our overall findings. At a finer resolution of 30 meters, FABDEM[53] was also considered but was ultimately found to contain building heights that skewed the terrain elevations. We delineate the urban region based on the administrative boundaries of each city.

### Synthetic urban forms
We conceptualize city urban form with rectangular landscape panels of length ($L$), width ($W$), slope ($\alpha$) and Manning coefficient ($n_M$) populated with $N$ square buildings of side length ($D$). Ensembles of urban landscapes representative of a specified porosity $\phi$ and order $\chi$ are generated through a hybrid reverse Monte Carlo (HRMC) algorithm, which was first introduced to improve upon the modeling of amorphous carbon structures[35,36]. Using user-defined constraints to inform the placement of buildings and an energy penalty to avoid unrealistic or physically improbable configurations, this probabilistic and iterative algorithm is capable of generating ensembles of urban forms with the targeted porosity and order (See Supplementary Note II and Supplementary Fig. S1 for additional detail).

Urban landscape panels were constructed with a length $L$=10,000 m, $W$=500 m, building size $D$=15 m, and a range of slopes, $\alpha$=0.001-0.1. The channel was configured sufficiently long to balance gravitational effects and flow resistance from bottom shear and form drag, sufficiently wide to avoid edge effects, and sufficiently small to make fine-resolution modeling feasible over thousands of scenarios. Manning's roughness ($n_M$) is 0.02 s/m$^{1/3}$ in all simulations, representing a concrete surface, and an ensemble of $M$=3 realizations are utilized for each combination of porosity, order, and slope. We confirmed that a set of three simulations was sufficient for estimating the mean flood depth by checking sixteen different urban forms using $M$=30 realizations and seeing only a negligible difference.

### Flow simulation
Flood flow for each realization of urban form is simulated using Par-BreZo, which uses a Godunov-type, finite volume scheme to solve the full two-dimensional shallow-water equations on an unstructured grid of triangular cells constrained by building walls[23]. Hence, buildings are represented by the so-called building-hole method[28].

Every realization of urban form is independently meshed through constrained Delaunay triangulation using the Triangle mesh generation software[54]. To ensure a quality mesh, all cell angles are restricted to greater than 20°, and a spatially-dependent maximum area constraint on cell size is utilized. Within the upstream section of the urban form where we seek to achieve a uniform flow in which gravity and flow resistance balance each other, i.e., 0 < $x$ < $L/D$ = 33.3, the maximum area constraint is $A_c/D^2$ = 0.02, where $A_c$ is the maximum area of a triangular cell. Hence, we use a cell size that is no larger than 2% of the size of a building. In the downstream portion of

the urban form, i.e., $L/D = 33.3 < x < L/D = 666.7$, the mesh coarsens to a maximum cell area of $A_c/D^2 = 0.13$ to reduce computational costs without impacting precision. See Supplementary Fig. S2 for a visual of the mesh.

To capture uniform flow within the upstream portion of each urban form, a flow rate of $Q$ is evenly spread over the upstream edge of the domain, the downstream edge is treated as a free-overfall boundary condition, and the sides of the domain are treated as free-slip walls. The model is initialized with an initially dry floodplain and integrated into a steady state with an adaptive time step that maintains a Courant number of 0.8. Upon convergence, the uniform depth and velocity distribution are evaluated based on the flow conditions within the upper 100 m of the channel. For each model, the urban form and flow attributes $\phi$, $\chi_{c_n}$, $Q$, $\bar{l}_c$, and $\alpha$, along with the resulting flow height $h$, velocity $u$, and their product $hu$, are provided in the spreadsheet 'Supplementary Data 1'.

We note that the domain length and width specification, grid refinement criteria, and convergence conditions were checked from an uncertainty perspective, and sensitivities were found to be negligible. Moreover, our multi-step simulation workflow, including (1) construction of synthetic urban forms, (2) mesh generation, (3) shallow water simulation, and (4) data post-processing, is implemented as a high-throughput framework. This framework runs on a high-performance computing cluster and streamlines the execution of parametric studies such as this one; see Supplementary Note III for details.

### Global urban flood hazard analysis

Each of the 20 cities identified for this study is decomposed into 1 km x 1 km panels, Fig. 5a. While many cities have street drains with O(100m) spacing to collect rainwater, these systems are only designed to contain flows from relatively frequent events (e.g., ten-year return period) and thus substantially longer overland flow runs occur with the most severe events. The use of a value of 1 km allows us to resolve both the amount of overland flow and the slope of the ground, thus providing an assessment of urban flooding at the neighborhood scale. Moreover, we find that the trends in flood hazards revealed by the mean-flow theory do not depend on a specific value of the rainfall accumulation length scale. Scarcely developed regions with a porosity higher than 90%, and within city boundaries, are disregarded. For each panel, urban porosity and average building side length are calculated from the building footprint data. Additionally, the DEM of each panel is analyzed to find the maximum slope and direction. We subsequently calculate the effective chord length in the direction of maximum slope in each cell, Fig. 5b and Supplementary Table S1 provide the complete set of urban attributes computed for the twenty cities studied in this work.

The Emergency Events Database (EM-DAT)[45] is cross-referenced for "flash flood events" and "tropical cyclones" since 2000, which lead to direct runoff from impervious surfaces into the urban form and the concentration of flooding. A total of sixteen flash flood events are identified, and five of these contain estimates of monetary damages, see Supplementary Table S2. Additionally, a total of seven tropical cyclone events are identified and all contain information about monetary damages. For each event, we collect precipitation data from the PERSIANN system[55]. Precipitation data is reported hourly for the extent of each event, and the highest rate is applied to the city area to produce flow rate $Q$ using the rational method and assuming that all precipitation becomes runoff (i.e., no interception or infiltration).

We do not consider flooding events that occurred before the year 2000 because PERSIANN satellite data for systematically estimating precipitation is not available, and because of substantial urban growth over the past twenty years. We also note that monetary losses recorded in the EM-DAT database are usually not confined to a city's boundaries

but include all regions affected by the event. This may cause an over- or under-estimation of damages reported for a city. We also note that monetary losses from tropical cycles may be caused by multiple hazard drivers including wind, storm surge, and rainfall, whereas monetary losses from flash floods are mainly the result of intense rainfall[2].

## Data availability

The results of the flow simulations through synthetic urban forms of varying porosity and order, presented in Fig. 4, generated in this study are provided in the spreadsheet 'Supplementary Data 1'.

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

## Acknowledgements

M.J.A.Q. was partly supported by NSF CAREER Award No. 2145537 and NSF Award No. 2103125. B.F.S. was supported by NSF Award No. 2031535. S.B. was supported as a GAANN Fellow through the UCI Civil and Environmental Engineering GAANN grant funded by the US Department of Education (P200A210077). S.B. also gratefully acknowledges the Bridge Fellowship from UCI's Henry Samueli School of Engineering.

## Author contributions

B.F.S. and M.J.A.Q designed the research. S.B. developed the high-throughput computational framework for the parametric study, performed numerical simulations, and gathered the simulation results. S.B. collected and assimilated urban, precipitation, and disaster data from multiple public sources. M.J.A.Q and B.F.S. developed the theoretical framework based on dimensional analysis. All authors analyzed the results and wrote the manuscript.

## Competing interests

The authors declare no competing interest.
