## [Peer Review File · Nature Communications]

How Urban Form Impacts FloodingEditorial Note: Parts of this Peer Review File have been redacted as indicated to remove third-party material where no permission to publish could be obtained.

REVIEWER COMMENTS

Reviewer #1 (Remarks to the Author):

The paper presents an integration of Boltzmann theory from thermodynamics/statistical mechanics with hydraulic theories, offering a new approach for understanding urban flood hazards. The approach tries to bridge the gap between the microscopic properties of complex urban environments and their macroscopic behaviors to create a proxy for characterizing flood vulnerability in urban settings. This is particularly significant for informing future urban development and planning, as it considers the intricate configuration of urban layouts - including building blocks and their arrangement - which are key factors in redistributing flood risk, as well as in altering flood inundation depths and velocities. The study's foundation is built upon a series of assumptions and an analytical approach that are both sound and scientifically rigorous. However, several minor aspects have been identified that require attention before publication to enhance the overall quality and clarity of the paper.

Specific comments:

1- Could you please ensure that the citation on line 94 precedes the period for consistency with citation formatting guidelines?

2- In line 96, it appears that 'k, j' should be revised to 'kj'.

3- In Figure 2, the white squares representing building blocks might be confused with the flood velocity color bar, where white indicates maximum velocity. To avoid this confusion, consider reversing the color bar so white represents minimum velocity or zero (the velocity in at building location is zero). Additionally, explicitly stating that white squares signify building blocks or voids in the flood model domain would enhance clarity. For a more direct comparison, displaying figures with the same porosity for square- and hexagon-like patterns (instead of 0.6 and 0.8) would be beneficial. Could you clarify the choice of showing a square pattern with porosity 0.8 and a staggered pattern with porosity 0.6? Also, adding a flow direction arrow to this figure would be helpful.

4- In line 119, it would be beneficial to define 'u' and 'Q'.

5- Lines 137-139 suggest a bimodal distribution for the staggered pattern, but it appears more unimodal. Could this be clarified?

6- In line 140, 'P(l'c)' should be revised to 'l'c'. The statistical expectation is defined over the random/state variable rather than its probability.

7- The definition of 'unit cells' in the manuscript seems unclear. Could you specify whether it refers to the length of the flood model domain or a segment of it?

8- In Figure 3, panel b, the two green lines are challenging to distinguish from each other. Perhaps a slight adjustment in the color resolve this.

9- In Figure 3, panel a, 'h' needs to be displayed perpendicular to the sloped surface. The gravitational force calculation uses α instead of $\sin(\alpha)$, suggesting an assumption of an insignificant slope. However, considering that a slope of 0.1 or 10%, as shown in panel c, appears quite steep, this approximation might not be accurate. Could the authors please provide their perspective or rationale on this matter?

10- In Equation 3, 'CD' needs to be square rooted; or using a different nomenclature, such as 'CD'.

11- In line 191, 'cn' is defined, but it would be more reader-friendly to introduce this definition at its first occurrence, such as in Equation 1. Applying this approach throughout the manuscript would

greatly aid reader comprehension.

12- In the Supplementary Information, the 'q' in Equation 11 seems like it should be changed to 'q2'

13- Figure 7 in the Supplementary Information is challenging to interpret due to the added lines that create a 3D perspective effect.

14- Regarding the chord length calculation, is it intended to be in any direction, or specifically in the dominant slope or flow direction? While it seems that you calculated it in the dominant slope direction, a general clarification on this approach would be insightful.

Reviewer #2 (Remarks to the Author):

I found the manuscript entitled "How Urban Form Impacts Flooding" by Balaian et al to be a very interesting and well written. The authors propose to my knowledge a unique method "mean-flow theory" to explore the effect of city structure on depth-velocity, an important factor for estimate damage.

I am not a mathematician but I found not issues with the description of their proposed methodology. I do believe that this work can have a significant impact in the area of flood analysis and risk; however, I do have a number of comments and questions on the methodology and some of the choices made. These are not presented in any order of seriousness.

1) The urban forms presented in Figure 2 and used to developed the data presented in Figure 4 assume that either the building forms are either uniform or hexagon-space and that they can be distributed randomly across a domain. While this might be the case for purposely build city (e.g most North American cities), I question if this assumption is valid elsewhere as cities usually grow organically. This is a major assumption that is not fully addressed.

2) While I understand the need to show the validity of the numerical approach, hence the use of synthetic data. I would like to have seen the same approach applies to a range of cities across the globe to see if the same relationship are valid in the main manuscript and not in the supplementary data. The analysis and discussion in section III is interesting and probably more relevant to readers than the synthetic data.

3) In the section III, line 248, a 1km length scale was chosen for runoff accumulation. How was this value selected? This also seems a large value in an urban setting.

4) The MERIT DEM was used in this study to obtain slopes. While MERIT is an improvement over STRM there is still errors in urban area with an average RMSE $\sim 4\text{m}$ (Liu et al 2021). Also it has a footprint of approximately $\sim 90\text{m}$ so approximately 100 pixels in each 1km grid. Is it possible the slopes calculated might not represent the terrain but a surface (building height), what would the impact of this be on the results.

Dear Editor and Reviewers,

We would like to thank you for reviewing our submission, validating the originality and significance of our contribution, and responding with constructive comments and questions. The authors have reviewed these queries and identified and implemented a number of minor revisions which we believe are responsive to these points, and further strengthen the paper. We have also prepared a point-by-point response to each query.

In the narrative that follows, text is formatted such that **reviewer comments appear in black**, **author responses appear in blue**, and **changes to the manuscript appear in red**.

We hope you find our responses, modification, and clarifications satisfactory, and we look forward to receiving your further comments and decision.

On Behalf of all authors,
MJ Qomi

REVIEWER COMMENTS: Reviewer #1

The paper presents an integration of Boltzmann theory from thermodynamics/statistical mechanics with hydraulic theories, offering a new approach for understanding urban flood hazards. The approach tries to bridge the gap between the microscopic properties of complex urban environments and their macroscopic behaviors to create a proxy for characterizing flood vulnerability in urban settings. This is particularly significant for informing future urban development and planning, as it considers the intricate configuration of urban layouts - including building blocks and their arrangement - which are key factors in redistributing flood risk, as well as in altering flood inundation depths and velocities. The study's foundation is built upon a series of assumptions and an analytical approach that are both sound and scientifically rigorous. However, several minor aspects have been identified that require attention before publication to enhance the overall quality and clarity of the paper.

We thank the reviewer for carefully reading the paper, summarizing our work and its significance, and offering constructive suggestions for improvements. Each of these comments and all reviewer questions have been considered as documented below.

Specific comments:

1- Could you please ensure that the citation on line 94 precedes the period for consistency with citation formatting guidelines?

This is addressed in the revised manuscript.

2- In line 96, it appears that 'k, j' should be revised to 'kj'.

This question refers to the angle notation in the calculation of the Mermin order parameter. As written in the old version of the manuscript: " $\theta_{kj k_0}$ is the angle between k , j , and k_0 buildings' centroids and k_0 is a fixed neighbor." This means the angle is calculated

between 3 buildings with indices k , j , and k_0 , where j is the central building and k_0 is a random neighbor fixed in the vicinity of j , with respect to which we calculate the angle $\theta_{kj k_0}$. By doing this we can remove the orientational dependence of the order parameter so that it becomes rotation invariant. To make this point clear in the revised manuscript, we have applied the following edit:

“... the central building j and its neighbors k and k_0 , where k_0 is a fixed neighbor and k permutes over all possible neighbors of the j th building.”

3- In Figure 2, the white squares representing building blocks might be confused with the flood velocity color bar, where white indicates maximum velocity. To avoid this confusion, consider reversing the color bar so white represents minimum velocity or zero (the velocity in at building location is zero). Additionally, explicitly stating that white squares signify building blocks or voids in the flood model domain would enhance clarity. For a more direct comparison, displaying figures with the same porosity for square- and hexagon-like patterns (instead of 0.6 and 0.8) would be beneficial. Could you clarify the choice of showing a square pattern with porosity 0.8 and a staggered pattern with porosity 0.6? Also, adding a flow direction arrow to this figure would be helpful.

This is an especially good point. Thank you. To improve Figure 2, we have changed the color of the buildings to gray and stated in the caption that gray squares represent flow obstructions. We have also considered your comment and have replaced the Figs. 2(a-c) and (g-i) with those of 0.6 porosity. As we observe, the qualitative results do not change, and conclusions remain the same.

4- In line 119, it would be beneficial to define 'u' and 'Q'.

The original manuscript reads as “Figs. 2(a-f) show the color map of non-dimensionalized flood velocity in the downslope direction, uD^2/Q , for neighborhoods of varying porosity and disorder.” We have changed the manuscript to address the undefined variables as follows:

“Figs. 2(a-f) show the color map of non-dimensionalized flood velocity in the downslope direction (uD^2/Q) as a function of flow velocity (u), flow rate (Q), and building size (D), for neighborhoods of varying porosity and disorder.”

5- Lines 137-139 suggest a bimodal distribution for the staggered pattern, but it appears more unimodal. Could this be clarified?

The chord lengths for perfectly staggered patterns are bimodal, but both modes feature a small l_c/D (<5) and the first mode is difficult to see in the original graphic. We have updated the graphic with a small inset to better show the two modes. Please see the updated manuscript.

6- In line 140, 'P(l'c)' should be revised to 'l'c'. The statistical expectation is defined over the random/state variable rather than its probability.

Correct. This is corrected in the revised manuscript.

7- The definition of 'unit cells' in the manuscript seems unclear. Could you specify whether it refers to the length of the flood model domain or a segment of it?

This is a very good question. To remove the channel size effects on the asymptotic value of flood depth in drawdown profiles, we needed to extend our channel length to 10 km. Ideally, we wanted to forgo with defining a unit cell and populate the 10 km channel with buildings at a given porosity and order parameter. However, we could not do this due to the limitations of the Hybrid Reverse Monte Carlo (HRMC) approach. In fact, it was computationally unfeasible to converge HRMC for a 10 km channel. To resolve this computational limitation, we had to define a unit cell, by repetition of which we could generate our 10 km channels. The size of this unit cell has two constraints: 1) it should be much larger than the building size, and 2) it should be greater than the chord length commonly found in urban forms across the world. As demonstrated in Figure 5c, the maximum average chord length in the twenty cities we studied belongs to São Paulo, which is ~12 times the average size of buildings. We also note that the standard deviations are not too large and in the order of a couple of building length. With this length and computational cost in mind, we chose a unit cell which is ~33 times the size of a building. We revised both the manuscript and SI to reflect this discussion:

Main Text: “Due to HRMC's computational cost, we only generate building patterns in a unit cell of length l_u and repeat the unit cell to cover the entire channel length.”

Caption of Figure 2: “While the unit cell l_u is ~33D, the channel length is set to $20l_u$ to mitigate size effects.”

In the SI, we have also provided the excerpt of this discussion.

8- In Figure 3, panel b, the two green lines are challenging to distinguish from each other. Perhaps a slight adjustment in the color resolve this.

We have addressed this request in the revised Figure 3(b) in the manuscript.

9- In Figure 3, panel a, 'h' needs to be displayed perpendicular to the sloped surface. The gravitational force calculation uses α instead of $\sin(\alpha)$, suggesting an assumption of an insignificant slope. However, considering that a slope of 0.1 or 10%, as shown in panel c, appears quite steep, this approximation might not be accurate. Could the authors please provide their perspective or rationale on this matter?

Thanks for the comment. We defined h as the vertical flood height. We have re-written the equations in the revised manuscript to include the consideration of large slopes. As shown in the revised equation 3 in the manuscript, $h \propto 1/(\cos(\alpha) \sqrt{\sin(\alpha)})$. In the limit of small slopes, this can be approximated as $h \propto 1/\sqrt{\alpha}$. For a slope of 10%, $\alpha = 0.0997$, which yields $\cos(\alpha) \sqrt{\sin(\alpha)} = 0.3142$. The $\sqrt{\alpha}$ is equal to 0.3158, which means the approximation introduces $\sim 0.005\%$ error in our calculations. Given the uncertainties in shallow water modeling, this level of error can be safely neglected. To reflect this discussion, we have included the following sentence in the revised manuscript:

“For the maximum 10% slope considered in this study, the approximation of $\cos(\alpha)\sqrt{\sin(\alpha)} \propto \sqrt{\alpha}$ introduces less than 1% error.”

10- In Equation 3, 'CD' needs to be square rooted; or using a different nomenclature, such as 'CD'.

Thanks for pointing this mistake. We have corrected the CD to include square root in the revised manuscript.

11- In line 191, 'cn' is defined, but it would be more reader-friendly to introduce this definition at its first occurrence, such as in Equation 1. Applying this approach throughout the manuscript would greatly aid reader comprehension.

In the original manuscript line 191 we defined c_n as our coordination number and wrote “... and q_{c_n} are specific to square and hexagonal symmetries as denoted by the coordination number c_n .” And in line 93 we wrote “At the scale of individual particles (or buildings), the Mermin order parameter determines the symmetry of c_n nearest neighbors to the reference particle [34].” To improve the readability, we have added the following text to clarify what c_n means after its first appearance in the manuscript as follows:

“For instance, for building arrangements with a square symmetry, $c_n=4$, and those with hexagonal symmetry feature $c_n=6$.”

12- In the Supplementary Information, the 'q' in Equation 11 seems like it should be changed to 'q2'

The issue is addressed in the revised SI.

13- Figure 7 in the Supplementary Information is challenging to interpret due to the added lines that create a 3D perspective effect.

We have redone Figure 7 to make it easier to understand and interpret.

14- Regarding the chord length calculation, is it intended to be in any direction, or specifically in the dominant slope or flow direction? While it seems that you calculated it in the dominant slope direction, a general clarification on this approach would be insightful.

This is an important point. In fact, the flood height is a function of the chord lengths in all directions, *i.e.*, $h = h(l_c(\theta))$, where θ is the angle with the steepest descent. In mathematical language, h is a functional of l_c . Although it is intuitive that a theoretical framework should consider functional formulation, the construction of such theory becomes increasingly involved. Therefore, in this paper we simplify this concept and only focus on chord lengths in the steepest descent. To reflect this discussion, we have added the following sentence to the manuscript:

“To characterize how the symmetry and irregularity interact with ground slope to control the neighborhood-scale flood levels, we leverage the non-dimensionalized chord length distribution $P(l'_c(\theta))$, which represents the distribution of randomly placed line segments fitting within pore space between two building walls in all directions (θ), see SI Appendix, Note SIV and Fig. S4 for detail. For simplicity, we here focus on chord length in the direction of steepest descent.”

REVIEWER COMMENTS: Reviewer #2

I found the manuscript entitled "How Urban Form Impacts Flooding" by Balaian et al to be a very interesting and well written. The authors propose to my knowledge a unique method "mean-flow theory" to explore the effect of city structure on depth-velocity, an important factor for estimate damage. I am not a mathematician but I found no issues with the description of their proposed methodology. I do believe that this work can have a significant impact in the area of flood analysis and risk; however, I do have a number of comments and questions on the methodology and some of the choices made. These are not presented in any order of seriousness.

We thank the reviewer for carefully reading the paper, summarizing our work and its significance, and offering construction suggestions for improvements. Each of these comments and all reviewer questions have been considered as documented below.

1) The urban forms presented in Figure 2 and used to developed the data presented in Figure 4 assume that either the building forms are either uniform or hexagon-space and that they can be distributed randomly across a domain. While this might be the case for purposely build city (e.g most North American cities), I question if this assumption is valid elsewhere as cities usually grow organically. This is a major assumption that is not fully addressed.

This is an important question. Indeed, our reduced-complexity model of urban form was configured with a Mermin "order" parameter for the purpose of approximating both "planned" and "organic" urban forms. Synthetic urban patterns with high order ($\chi_6 > 0.9$ and $\chi_4 > 0.9$) can be viewed as "planned" urban forms, similar to those observed in North America, e.g., Chicago and Seattle. Urban forms with low order ($\chi_6 < 0.7$ and $\chi_4 < 0.7$) can be viewed as organically formed and evolved urban environments, e.g., Jakarta, Indonesia and Savannah, GA, USA. To reflect this discussion, we have made the following modification to the revised manuscript:

"When χ_{c_n} is greater than 0.9, the synthetic urban form exhibits a pronounced symmetry and resembles a planned urban layout, e.g., Chicago, Fig. 1(c). As disorder increases, the symmetry diminishes and the synthetic urban form features characteristics of an organically nucleated and grown urban environment, e.g., Jakarta, Fig. 1(e)."

2) While I understand the need to show the validity of the numerical approach, hence the use of synthetic data. I would like to have seen the same approach applies to a range of cities across the globe to see if the same relationship are valid in the main manuscript and not in the supplementary data. The analysis and discussion in section III is interesting and probably more relevant to readers than the synthetic data.

Given journal specifications for the maximum word count and display items, we are unfortunately unable to expand the content in main text. Our decision to prioritize the synthetic urban forms for presentation in the main text stems from its crucial role in establishing the universality of the proposed mean flow theory for urban flooding.

3) In the section III, line 248, a 1km length scale was chosen for runoff accumulation. How was this value selected? This also seems a large value in an urban setting. We appreciate this question and agree that this selection was not adequately justified in our first submission.

This distance (1000 m) represents an order-of-magnitude estimate for the distance travelled by rainfall runoff during extreme rainfall before finding a primary drainage channel. Many cities are developed with urban drainage systems that include curb inlets with $O(100\text{ m})$ spacing, but these systems are usually only designed to capture rainfall with a return period of 10 or 20 years at most. Hence, rainfall runoff must run over substantially longer distances during rainfall events with larger return periods. At the upper end, rainfall runoff over a distance of $O(10\text{ km})$ was viewed as highly unlikely due to the typical distribution of urban drainage channels that intercept overland flow.

We note that our results are not especially sensitive to the precise distance used for this calculation, since our results are collapsed through dimensional analysis. In the following, we observe Figures 4(e-f) with the original 1km runoff length (a&c), and with a 100 m runoff instead (b&d). As we can see, this change in runoff length does not affect our qualitative findings, but merely scales our values due to the reduced runoff.

Another benefit of using a length scale of 1 km is that it is a reasonable spatial scale to capture the slope of a neighborhood, and generally aligned well with the availability of global elevation data required to estimate slopes.

The paper has been revised to clarify our justification for the length scale of rainfall accumulation, as follows.

“While many cities have street drains with $O(100\text{ m})$ spacing to collect rainwater, these systems are only designed to contain flows from relatively frequent events (e.g., 10 yr return period) and thus substantially longer overland flow runs occur with the most severe events. Use of a value of 1km allows us to resolve both the amount of overland flow and the slope of the ground thus provides an assessment of urban flooding at the scale of neighborhood. Moreover, we found that the trends in flood hazards revealed by the model were not dependent on a specific value of the rainfall accumulation length scale (see methods).”

4) The MERIT DEM was used in this study to obtain slopes. While MERIT is an improvement over STRM there is still errors in urban area with an average RMSE $\sim 4\text{m}$ (Liu et al 2021). Also it has a footprint of approximately $\sim 90\text{m}$ so approximately 100 pixels in each 1km grid. Is it possible the slopes calculated might not represent the terrain but a surface (building height), what would the impact of this be on the results.

We appreciate the reviewer’s comments about the error in the MERIT dataset. Since we considered a unit cell of 1 km x 1 km, this error can translate to an additional ± 0.008 change in the slope. We have included this error in the revised manuscript and SI. This error would most likely be present on all cities and therefore has little effect on our overall findings, as the difference between the highest (San Francisco) and lowest (Houston) slope is ~ 0.18 , which is much greater than ± 0.008 error introduced by the inaccuracy in the DEM.

We also note that the accuracy of slopes was a major consideration in this study. Initially, we worked with two main DEMs, namely FABDEM (30m resolution) and MERIT (90m resolution). We found that although FABDEM claims to remove forests and buildings from DEMs, it behaves poorly in dense urban regions populated with tall buildings. For example, the FABDEM data in Manhattan shows spuriously high elevations in Midtown and Lower Manhattan (see below). These issues appear to be resolved in the MERIT DEM for these regions of Manhattan, although at the expense of lower resolution. Nevertheless, since we seek slopes at the scale of 1 km, we feel that MERIT offers adequate spatial resolution.

[redacted]

The following has been added to the revised manuscript to reflect this discussion:

“While the MERIT DEM was found to contain a RMSE of ~4m (Liu et. al.), this error translates to a ± 0.008 change in slope and is likely present in all cities, having little effect on our overall findings. At a finer resolution of 30 meters, FABDEM (Hawker et. al.) was also considered but was ultimately found to contain building heights which skewed the terrain elevations.”

REVIEWERS' COMMENTS

Reviewer #1 (Remarks to the Author):

The authors have done a good job in addressing all of my previous comments. I don't have any additional comments.

Reviewer #2 (Remarks to the Author):

Dear Authors,
Thank you for addressing my comments.